# iCanClean Removes Motion, Muscle, Eye, and Line-Noise Artifacts from Phantom EEG

**DOI:** 10.3390/s23198214

**Published:** 2023-10-01

**Authors:** Ryan J. Downey, Daniel P. Ferris

**Affiliations:** J. Crayton Pruitt Family Department of Biomedical Engineering, University of Florida, Gainesville, FL 32611, USA; downeyryanj@gmail.com

**Keywords:** EEG, noise cancellation, artifact removal, motion artifacts, muscle artifacts, phantom head

## Abstract

The goal of this study was to test a novel approach (iCanClean) to remove non-brain sources from scalp EEG data recorded in mobile conditions. We created an electrically conductive phantom head with 10 brain sources, 10 contaminating sources, scalp, and hair. We tested the ability of iCanClean to remove artifacts while preserving brain activity under six conditions: *Brain*, *Brain + Eyes*, *Brain + Neck Muscles*, *Brain + Facial Muscles*, *Brain + Walking Motion*, and *Brain + All Artifacts*. We compared iCanClean to three other methods: Artifact Subspace Reconstruction (ASR), Auto-CCA, and Adaptive Filtering. Before and after cleaning, we calculated a Data Quality Score (0–100%), based on the average correlation between brain sources and EEG channels. iCanClean consistently outperformed the other three methods, regardless of the type or number of artifacts present. The most striking result was for the condition with all artifacts simultaneously present. Starting from a Data Quality Score of 15.7% (before cleaning), the *Brain + All Artifacts* condition improved to 55.9% after iCanClean. Meanwhile, it only improved to 27.6%, 27.2%, and 32.9% after ASR, Auto-CCA, and Adaptive Filtering. For context, the *Brain* condition scored 57.2% without cleaning (reasonable target). We conclude that iCanClean offers the ability to clear multiple artifact sources in real time and could facilitate human mobile brain-imaging studies with EEG.

## 1. Introduction

Electroencephalography (EEG) is an effective tool for non-invasively recording brain activity. Compared to functional brain-imaging modalities which record relatively slow blood-oxygen-level-dependent signals (e.g., fMRI, fNIRS), EEG records electrocortical dynamics with high temporal resolution. Additionally, in high-density applications (roughly 100+ channels), EEG also has the ability to localize cortical sources with reasonable spatial resolution (<1 cm; see Table 2 in [1] and Figure 4A in [2]), and even subcortical activity can be reconstructed from scalp EEG and localized (approximately 2 cm; see Table 1 in [3]). Last, EEG is both portable and relatively cheap, making it ideal for studying the neural control of whole-body movement, both in laboratory settings [4,5,6] and in the natural environment [7,8].

One drawback with EEG is that it is often contaminated by a wide variety of artifacts. Depending on the recording environment, the EEG equipment being used, the task being studied, and individual variation in each participant’s behavior, a wide variety of artifacts hinder the ability to isolate electrocortical sources. Some artifacts are internal (or biological) in origin, such as muscle contractions [9,10,11], eye blinks [12], and eye saccades [13]. Other artifacts come from the external environment, such as line-noise artifacts and interference from various items of electrical equipment. Additional artifacts originate from more complex mechanisms, for example, motion artifacts, which are largely the result of cable sway [14]. As EEG cables sway through the air, they interact with each other and background electromagnetic fields. Inductive coupling and electromagnetic radiation in the environment can have a relatively large effect on EEG signal quality given the small voltage of electrocortical signals as they appear at the scalp (roughly 20 µV). Active electrodes can mitigate motion artifacts by amplifying the EEG signals prior to transmission and digital sampling. However, motion artifacts are still problematic for recording high-fidelity EEG signals, especially when participants are moving around in space and activating many muscles about their neck and face.

There are many software approaches to removing EEG artifacts (see [15]), but no single method reliably cleans all types of artifacts and many methods are not well suited for real-time implementation. Independent component analysis (ICA) is one of the most common approaches for blind EEG source separation and artifact removal. ICA can effectively remove eye blinks, muscle, and line noise [16,17]. It can also extract high-quality, independent brain sources from mixed EEG data. However, ICA is generally computationally slow. Computation time varies by application and algorithm used, but in our experience with high-density (100+ channel) EEG, the Infomax ICA algorithm can easily require 5+ h of computation time on a modern home computer or work computer (non-supercomputer) to decompose less than an hour of data. On a supercomputer with 64 CPU cores, it takes approximately 1 h to decompose 48 min of mobile EEG data with the AMICA algorithm. Further, ICA generally requires a large amount of data to be recorded to ensure a good decomposition (see Section 3.5 of [18] for additional detail). Specifics vary by application, but our general recommendation for mobile scenarios is to record at least 30 min of high-density EEG (100+ channels) at a sampling frequency of at least 500 Hz when attempting to separate sources with ICA. Generally speaking, ICA is not well-suited for real-time cleaning, although attempts have been made to remove eye blinks in real time with ICA [19]. Similarly, ensemble empirical mode decomposition [20,21] is not further discussed due to its high computational cost.

When reference noise recordings are available, Adaptive Filtering is a popular approach for removing artifacts in real time [22]. Adaptive Filtering works by scaling one or more reference noise signals to optimally fit onto a single corrupted data signal of interest (minimizing the mismatch) and then subtracting the best fit [23]. Although often implemented recursively (online), Adaptive Filtering is based on the concept of linear regression, which can easily be solved (offline). Adaptive Filtering has been used to remove eye artifacts [23,24]. Eye artifacts, in particular, are well suited for Adaptive Filtering because it is relatively easy to obtain accurate recordings of eye artifacts with only 2–4 electrooculogram electrodes. One limitation of Adaptive Filtering, however, is that it assumes noise sources project onto the EEG sensors and noise sensors nearly identically, with no significant difference other than simple scaling (linear mixing). In certain scenarios, Adaptive Filtering may exhibit poor performance and need to be modified. For example, in [25] the authors reported poor performance when they tried to remove EEG motion artifacts using an inertial measurement unit (IMU) attached to the head (reference noise signals) and traditional Adaptive Filtering. To achieve better performance, the authors modified the approach by using Volterra series expansion terms to capture nonlinear effects in the IMU signals. However, to implement the approach in [25], users must a priori know the frequency of the motion artifact and assume it remains the same throughout the data collection. Thus, the approach in [25] may not be easily generalizable across tasks, types of artifacts, or types of noise sensors.

When reference noise recordings are not available, Artifact Subspace Reconstruction (ASR) and Auto-Canonical Correlation Analysis (Auto-CCA) are two popular options. ASR is included by default with EEGLAB as an offline cleaning option, but it was first developed for real-time cleaning as part of BCILAB [26,27,28]. The burst repair feature of ASR is based on principal component analysis. ASR has been shown to help with muscle and eye artifacts [29]. ASR does not require reference noise signals; however, it does require clean EEG data for calibration. The user can supply the calibration data themselves (e.g., by recording clean resting-state data prior to recording the task of interest), or they can choose to automatically extract calibration data from contaminated data (assuming clean segments exist). Canonical correlation analysis (CCA) is a computationally efficient statistical method that can be used for blind source separation. See [30] for a review of the mathematics behind CCA and some suggested applications for neuroscientists. Researchers previously applied CCA to EEG data and a slightly lagged (but otherwise identical) EEG dataset [20,31,32]. This approach is often referred to as 1-sample lag CCA, but we prefer the term Auto-CCA (auto-correlation extended to CCA). Auto-CCA was developed based on the idea that a small (e.g., 1-sample) shift represents a negligible phase change for low-frequency content but a significant phase change for high-frequency content. Therefore, relatively low-frequency Auto-CCA components should have strong correlation while relatively high-frequency Auto-CCA components should have weak correlation. Auto-CCA was shown to remove high-frequency muscle artifacts in [20,31,32]. Theoretically, Auto-CCA can also remove low-frequency artifacts such as motion artifacts and eye artifacts by rejecting high-correlation components, but users should exercise caution to avoid accidental deletion of brain activity (also low frequency/high correlation).

Although ASR, Auto-CCA, and Adaptive Filtering are useful, there is still a need for improved cleaning performance and the ability to remove all types of artifacts with a single approach. We recently developed a novel generalized framework for removing EEG artifacts, termed iCanClean [33]. Compared to other real-time-capable methods, iCanClean is an all-in-one cleaning solution that does not require accurate reference recordings (see Adaptive Filtering), does not require clean example data for calibration (see ASR), and does not strongly risk removing brain activity (see Auto-CCA). Furthermore, as we will later demonstrate, iCanClean consistently outperforms ASR, Auto-CCA, and Adaptive Filtering, regardless of the type or number of artifacts present. Although we previously released the mathematics behind the iCanClean algorithm as a preprint [33], we have yet to demonstrate its ability to clean a wide variety of artifacts and quantitatively compare it to competing methods in a large parameter sweep. The objective of this work was to validate iCanClean on a phantom head apparatus, with known ground-truth brain signals, and to quantitatively compare iCanClean with other real-time-capable algorithms.

The outline of the rest of the paper is as follows. We first describe our novel framework for removing artifacts given reference noise signals. We then show how corrupt EEG signals can be temporarily converted to pseudo-reference noise signals to clean artifacts more generally (without the need for separate noise sensors). Using an electrical phantom head with embedded brain source antennae (i.e., ground-truth brain signals are available), we collected EEG data corrupted by motion, muscle, eye, and line-noise artifacts from the phantom. We then performed multiple parameter sweeps on the EEG data and compared our newly developed method (iCanClean) to three alternative methods (ASR, Auto-CCA, and Adaptive Filtering). After validating iCanClean on phantom EEG data (primary objective), we end with remarks on optimal settings for various scenarios and suggestions for future research directions. Specifically, we provide preliminary evidence that iCanClean can remove motion artifacts from human EEG data in real time using dual-layer EEG sensors. We also briefly explore the potential for iCanClean to find brain components from mixed EEG data, similar to ICA, but more computationally efficient (results within seconds or minutes instead of hours) and less total data required (5 min collections versus 30–45 min).

## 2. The iCanClean Algorithm

The acronym iCanClean stands for implementing Canonical correlation to Cancel Latent Electromagnetic Artifacts and Noise. iCanClean consists of four main steps, as depicted in Figure 1A. The first step is to identify candidate noise components by sending corrupt EEG data and reference noise data to canonical correlation analysis (CCA). CCA finds and returns, in ranked order, subspaces of the corrupt EEG channels that are most correlated with subspaces of the reference noise channels. The second step is to select a subset of noise components for removal, for example, those with the strongest correlation. The third step is to calculate the projection from the bad components to the EEG channels, for example, with linear regression. The fourth step is to directly subtract the projected noise components from the EEG channels. These steps can be applied to a large fixed window or to a smaller moving window (e.g., to deal with nonstationarity in the data). Additional mathematical detail is provided in the supplemental section titled “Mathematics of iCanClean” (File S1). When direct recording of reference noise signals with dedicated sensors is not possible, one alternative we developed is to make use of pseudo-reference noise signals. As depicted in Figure 1B, pseudo-reference signals can be created by taking contaminated EEG signals and applying a basic temporal filter to attenuate the majority of brain activity (e.g., a 5–45 Hz band-stop filter). Once the pseudo-reference signals are created, the rest of the algorithm is the same. We have packaged the iCanClean algorithm into an EEGLAB plugin so others can easily implement our method on their own data. The plugin includes a graphical user interface (see Appendix A), complete with the option for copying and/or filtering signals prior to finding noise sources in the EEG data (e.g., to create pseudo-reference noise signals). In addition to providing iCanClean as a downloadable EEGLAB plugin, we are also providing all relevant phantom EEG data and MATLAB scripts so others can replicate the results.

## 3. Methods

### 3.1. Phantom Head Apparatus

We used an electrical phantom head apparatus to validate iCanClean. This allowed us to have ground-truth signals to quantify the cleaning performance and to compare iCanClean with other real-time-capable cleaning algorithms. We made a custom phantom head (Figure 2), based on the open EEG phantom project [34]. The phantom head included antennae so that ground-truth electrical signals could be broadcast from inside the head; we used the tip (T) and sleeve (S) portion of 3.5 mm TRS audio jacks to create electric dipoles. There were 10 brain antennae, 4 neck-muscle antennae, 4 facial-muscle antennae, and 3 eye antennae (one for saccades, two electrically coupled for blinks). We mixed ballistics gelatin (1 kg), deionized water (5 L), and salt (50 g NaCl) and poured it into a plastic 3D-printed mold. The ballistics gelatin and salt mixture mimicked physical properties of human tissue and allowed for volume conduction of electrical signals. We added a wig (nonconductive) along with a layer of conductive fabric (EeonTex LTT-PI-100, Marktek Inc., Chesterfield, MO, USA) to mimic hair and scalp. We placed the phantom on a robotic motion platform (NOTUS, Symétrie, Nîmes, France) to induce motion artifacts [35,36].

### 3.2. Ground-Truth Brain Sources

We created ground-truth brain signals for the phantom head using neural-mass models [37,38,39]. We randomized the input parameters for 10,000 candidate neural-mass models, calculated the power spectral density (PSD) of the output waveform for each set of parameters, and we compared the power spectral profiles to ICA components coming from clean resting-state data (sitting eyes open). We kept the 10 neural-mass models whose PSD curves best matched human experimental data in the frequency band of 5–50 Hz (minimized sum-of-squares error). We broadcast the ground-truth brain signals into the phantom head with a digital-to-analogue converter (cDAQ-9178 chassis with NI 9269 output modules, National Instruments, USA). We scaled the signal coming out of the digital-to-analogue converter (going into the phantom) so that the EEG signals measured at the scalp were in physiological range (targeted 20 uV for the brain signals).

### 3.3. Ground-Truth Artifactual Sources

We created artifactual (contaminating) sources for the phantom head from example human data. Neck-muscle sources are the same as in [36], which were created from direct bipolar electromyography (EMG) recordings of the trapezius and sternocleidomastoid muscle during a 1.5 m/s walking task. Meanwhile, facial-muscle sources and eye-blink/saccade sources were newly recorded for this study from a single human subject. To create facial-muscle sources, we recorded EEG during a trial which consisted of chewing food, swallowing food, and drinking water. To create eye sources, we recorded EEG during frequent blinking. Unlike the neck-muscle sources, which were direct bipolar recordings, here we extracted example artifacts by sending mixed EEG data to independent component analysis. We broadcast known artifactual sources into the phantom head using a digital-to-analogue converter, targeting various amplitudes at the scalp (150 uV neck muscles, 300 uV facial muscles, 150 uV eyes).

### 3.4. Conditions Tested

We tested 6 conditions of different source mixes. The six conditions were: *Brain*, *Brain + Walking Motion Artifacts*, *Brain + Eye Artifacts*, *Brain + Neck-Muscle Artifacts*, *Brain + Facial-Muscle Artifacts*, and *Brain + All Artifacts* (i.e., all artifacts simultaneously present). The *Brain* condition represented relatively clean EEG data at rest, with no other artifacts purposefully introduced. The *Brain* condition, however, still contained trace amounts of 60 Hz line noise which could not be avoided. We broadcast muscle and eye artifacts into the phantom non-brain antennae. For motion artifacts (*Brain + Motion* and *Brain + All Artifacts*), we placed the phantom on the robotic platform to induce phantom movement matching walking head trajectory at 1.5 m/s [36]. For the conditions which did not involve motion (*Brain*, *Brain + Neck Muscles*, *Brain + Facial Muscles*, and *Brain + Eyes*), the platform was stationary with the motors turned off. For conditions where muscle and/or eye artifacts were not involved, the associated sources were not broadcast to the antennae (i.e., turned off electrically by setting the output gain to zero).

### 3.5. EEG Recording Apparatus

We recorded EEG with a custom-made 128 + 128 channel dual-layer scalp electrode system (ActiveTwo, BioSemi, Amsterdam, The Netherlands) [40]. The dual-layer system records raw noise signals (motion and line-noise artifact) alongside traditional EEG electrodes. We also recorded from 8 extra electrodes connected to the same EEG system. We placed these extra electrodes over the neck location on the phantom head, near the neck-muscle antennae locations underneath (2 per muscle; 1 superior, 1 inferior). We did not place extra electrodes over the facial-muscle antennae or eye antennae of the phantom because we prioritized having reference noise signals available for removing neck-muscle artifacts. To promote cable sway and increase motion artifact during the *Brain + Walking Motion* and *Brain + All Artifacts* conditions, we hung the electrode cables loosely. The digital-to-analog converter and EEG system were synchronized with trigger events at the beginning and end of each trial. This allowed us to directly compare the 10 ground-truth brain signals, which were broadcast into the phantom head, with the 128-channel EEG data recorded at the scalp. A video demonstration of the phantom head undergoing walking motion is provided in Video S4.

### 3.6. Parameter Sweep

We tested iCanClean against competing cleaning methods in a large parameter sweep (Figure 3). Raw EEG data were imported and high-pass filtered to remove large DC (0 Hz) offsets and slow drifts (*eegfiltnew* function, 1 Hz high pass = −6 dB at 0.5 Hz). No channels were rejected or re-referenced. After basic preprocessing, the data were sent to one of four methods: iCanClean, ASR, Auto-CCA, and Adaptive Filtering, with variations on each method and their parameters. We also tested an IMU-based Filtering approach for removing motion artifacts.

#### 3.6.1. iCanClean

For iCanClean, we varied two main parameters of interest. First, we varied the R^2^ threshold cutoff from 0 (max cleaning) to 1 (no cleaning). The R^2^ parameter adjusts the overall aggressiveness of the cleaning by determining how many noise components to remove (see File S1, Equation (2)). Second, we varied the type of reference noise signals. The choices varied between signals recorded by dual-layer EEG sensors (useful for removing motion and line-noise artifacts), signals recorded by neck-muscle electromyography (useful for removing neck-muscle artifacts), and pseudo-reference signals created from temporally filtered EEG (useful for removing artifacts when reference noise signals are otherwise not available). For demonstration purposes, we also tested the effect of temporarily re-referencing the EEG and noise signals prior to iCanClean (useful when noise signals are initially recorded with the same reference as the EEG signals). We tested cleaning the EEG data using mixtures of the EEG channels themselves (File S1, Equation (3)) versus using mixtures of the noise channels (Equation (3b)) versus using mixtures of both (Equation (3c)).

#### 3.6.2. Adaptive Filtering

For Adaptive Filtering, we varied an R^2^ threshold from 0 (max cleaning) to 1 (no cleaning) as well as the reference noise signal type, similar to iCanClean. The R^2^ threshold was used to determine which noise channels were sufficiently correlated to each EEG channel prior to performing linear regression. Typically for Adaptive Filtering, all noise channels would be included in the regression problem (i.e., always set R^2^ threshold = 0 for max aggressiveness). However, we implemented a range of thresholds to provide the fairest comparison possible (i.e, to demonstrate iCanClean’s improved performance is not due to the R^2^ thresholding aspect). For the *Brain + Walking Motion* condition, we also tested out an extension to Adaptive Filtering, described in [25], which cleans EEG data using accelerometer signals from an inertial measurement unit (IMU) attached to the head.

#### 3.6.3. Auto-CCA

For Auto-CCA, we varied the R^2^ threshold (0–1), the lag amount (1, 2, 3, or 4 samples at 512 Hz), and whether high- or low-correlation Auto-CCA components were removed. Note for Auto-CCA that R^2^ values of 0 and 1 can flip roles (max cleaning versus no cleaning), depending on the rejection setting (high/low correlation). In the literature, typically only low-correlation components are removed and only a lag of 1 is used. We also tested removing high-correlation components and tested multiple lags for completeness.

#### 3.6.4. ASR

For ASR, we varied the burst criterion value (1–250 standard deviations), which adjusts the overall aggressiveness of the cleaning. We also tested directly providing a clean dataset to ASR (external calibration) versus asking ASR to automatically find clean sections from the same data to be cleaned (auto calibration). For the external dataset, we provided ASR with clean, stationary data from the *Brain* (only) condition (i.e., the ideal scenario with no eye, muscle, or movement artifacts).

#### 3.6.5. IMU-Based Filtering

For IMU-based Filtering, we implemented an equivalent offline version of the algorithm described in [25], including the 2nd order Volterra expansion terms and cascade filtering approach. For the parameter sweep, we varied the filter bank frequencies (base frequency, 0.85–1.05 Hz; number of harmonics, 1–4) and the optimization type (minimize L_2_ norm versus L_infinity_ norm).

### 3.7. Quantifying Cleaning Performance (Data Quality Score)

We quantified the cleaning performance with a Data Quality Score, based on correlation between EEG channels and ground-truth brain sources. We first time-warped the EEG data to the ground-truth brain sources using event markers (start and stop of each trial). We then calculated Pearson’s R^2^ correlation matrix between the EEG channels and the ground-truth sources (128 EEG channels by 10 brain sources). We summed the R^2^ values across all ground-truth brain sources to calculate, for each EEG channel, the percentage variance which is explained by the brain sources. We then averaged across all channels to yield a scalar value that quantifies how well the EEG data can be explained by linear mixtures of the ground-truth brain sources (0–100%, higher is better). Note that a score of 100% can be achieved when all of the EEG channels are strictly mixtures of the 10 brain sources, with at least one brain source projecting onto any given EEG channel, and nothing else (no other extraneous sources to introduce noise). However, note that a score of 100% could also be achieved in the case where 1 or more brain sources are accidentally deleted during cleaning, so long as no artifacts remain on the EEG channels. Therefore, we introduced a correction factor (penalty term) to emphasize we do not want to accidentally delete brain activity during cleaning. The correction factor was calculated as follows. For each ground-truth brain signal, we used linear regression to first determine the best possible reconstruction that could be produced using linear mixtures of the minimally processed EEG data (Pre). After cleaning, we recalculated the best possible reconstruction for each ground-truth brain signal using mixtures of cleaned EEG data (Post). We quantified potential deletion of each ground-truth brain source using the ratio of the best reconstructions (variance accounted for Post divided by variance accounted for Pre). A ratio of 1 indicates no brain source removal, whereas a ratio of 0.75 would indicate 25% removal of brain activity (Post relative to Pre). After calculating the ratio for each of the 10 brain sources, we took the minimum value across all brain sources. We multiplied this correction factor to the raw score to calculate the final (corrected) data quality score.

### 3.8. Summarizing Results and Reproducability

We summarized the quantitative results for iCanClean, ASR, Auto-CCA, and Adaptive Filtering by comparing each algorithm’s best cleaning performance (i.e., Data Quality Score) for each condition. We then constructed time-series snapshots and power spectral density plots to qualitatively compare the algorithms at their optimal settings. We are providing all relevant data and scripts so others can replicate and build on our findings. Similarly, we are also providing the iCanClean algorithm as a downloadable plugin for EEGLAB, complete with a graphical user interface, to make it easier for others to test out the iCanClean algorithm on their own data.

## 4. Results

### 4.1. Summarized Quantitative Results (Main Takeaway)

We tested tens of thousands of parameter sets for iCanClean and competing methods (ASR, Auto-CCA, and Adaptive Filtering). We gathered the best outcome (best Data Quality Score) for each cleaning algorithm tested, condition by condition. As depicted in Figure 4, the iCanClean algorithm at its optimal settings consistently outperformed other real-time-capable EEG cleaning algorithms at their respective optimal settings. iCanClean achieved the highest score for all conditions except for the *Brain* condition (no purposefully imposed artifacts, only line noise present) where iCanClean placed second. iCanClean’s ability to clean the *Brain + All Artifacts* condition was particularly noteworthy (see highlighted yellow bow in Figure 4). Its Data Quality Score could not be matched by competing methods.

### 4.2. Qualitative Results (Supplementary Detail)

#### 4.2.1. Brain

All algorithms except ASR slightly cleaned the *Brain* condition, which had mild line-noise artifact contamination (unintended) but no purposefully imposed (experimentally manipulated) artifacts. This was the only condition where iCanClean was not the top performer, but iCanClean still performed well (second place) and the difference in the Data Quality Scores is negligible (59.1% iCanClean vs. 59.5% Adaptive Filtering; Figure 4). See Figure 5 for example time scrolls and PSDs before and after cleaning the *Brain* condition with each method. Note that Adaptive Filtering appeared to have removed some brain activity in addition to the line noise, as can be seen in the difference plot.

#### 4.2.2. Brain + Eyes

All algorithms improved the *Brain + Eyes* condition, but iCanClean (58.6%) and Auto-CCA (58.5%) were the best performers (Figure 4). See Figure 6 for example time scrolls and PSDs before and after cleaning the *Brain + Eyes* condition with each method. Note that Adaptive Filtering appears to have accidentally deleted some brain activity, as can be seen in the difference plot. Also note that the shape of the eye-blink artifacts that were removed are distorted (more sinusoidal) for Adaptive Filtering (see Difference plot around time = 2.5 s). This is due to Adaptive Filtering using band-stop filtered EEG channels as the pseudo-reference noise channels (no electrooculogram sensors/recordings available). Here, iCanClean used the same pseudo-reference noise channels as Adaptive Filtering, but iCanClean was able to extract a more accurate estimate of the eye-blink artifacts. Surprisingly, ASR did little to remove eye artifacts.

#### 4.2.3. Brain + Neck Muscles

All algorithms improved the *Brain + Neck Muscles* condition, but iCanClean (58.1%) and Adaptive Filtering (56.4%) were the best performers (Figure 4). See Figure 7 for example time scrolls and PSDs before and after cleaning the *Brain + Neck Muscles* condition with each method. Note that iCanClean was the only algorithm to simultaneously remove neck-muscle artifacts and 60 Hz line noise. All other algorithms left the line noise in place.

#### 4.2.4. Brain + Walking Motion

All algorithms improved the *Brain + Walking Motion* condition, but iCanClean (54.3%) clearly outperformed the rest (Figure 4; next best score = 43.3%). See Figure 8 for example time scrolls and PSDs before and after cleaning the *Brain + Walking Motion* condition with each method. Note that iCanClean not only removed motion artifacts but also 60 Hz line noise and other electromagnetic interference from the motors on the hexapod motion platform. Auto-CCA removed milder motion (low amplitude, consistent frequency) artifacts along with line noise (see Difference plot); however, Auto-CCA was not able to remove more severe artifacts (large amplitude fluctuations remain in the ‘Clean’ plot, especially prior to 2 s). Meanwhile, ASR was only responsive to cleaning the worst contaminated section (first 2 s) and largely was not helpful for removing motion artifacts. iCanClean was not sensitive to the sub-type of motion artifact; it performed well at all time points.

#### 4.2.5. Brain + Facial Muscles

All algorithms improved the *Brain + Facial Muscles* condition, but iCanClean (57.9%) and ASR (54.3%) were the best performers (Figure 4). See Figure 9 for example time scrolls and PSDs before and after cleaning the *Brain + Facial Muscles* condition with each method. Note that iCanClean not only removed facial-muscle artifacts but also 60 Hz line noise. Adaptive Filtering with pseudo-reference noise signals (band-stop filtered EEG) accidentally deleted brain activity, while iCanClean used the same pseudo-reference noise signals but kept brain activity intact.

#### 4.2.6. Brain + All Artifacts

When all artifacts were simultaneously involved, iCanClean offered a substantial improvement over all other methods. Starting from a score of 15.7% (before cleaning), the *Brain + All Artifacts* condition improved to 55.9% after iCanClean. Meanwhile, it only improved to 27.6%, 27.2%, and 32.9% after ASR, Auto-CCA, and Adaptive Filtering. For context, the *Brain* (only) condition scored 57.2% without cleaning, which is a reasonable value to target. See Figure 10 for example time scrolls and PSDs before and after cleaning the *Brain + All Artifacts* condition with each method. Note how the power spectra for the *Brain + All Artifacts* data (after iCanClean) resembles the already clean data in the *Brain* (only) condition (compare the blue curve in the last row of Figure 10 with the blue curve in the last row of Figure 5).

## 5. Discussion

### 5.1. Background and Objective

The objective of this work was to validate iCanClean, a novel method to identify and remove a wide variety of electroencephalography (EEG) artifacts, with or without reference noise signals. Compared to other popular cleaning algorithms, such as Adaptive Filtering, Auto-CCA, and ASR, our novel method has several theoretical advantages. Unlike Adaptive Filtering, which only considers one EEG channel at a time, iCanClean searches for latent relationships that exist between the entire set of EEG channels and the entire set of reference (or pseudo-reference) noise channels. iCanClean can identify noise components as they exist in the subspace of all EEG channels, similar to ICA. This should lead to better cleaning performance (i.e., better Data Quality Scores) in the scenario where noise sensors are imperfect or where the projection from the noise sources to the sensors differs between the EEG and the noise electrodes. Unlike Auto-CCA, which searches for relationships between EEG data and a lagged version of itself, iCanClean uses canonical correlation analysis (CCA) to search for relationships between EEG signals and reference (or pseudo-reference) noise signals. Thus, iCanClean exploits an extra set of valuable information not utilized by Auto-CCA, which should lead to better cleaning performance and reduce the likelihood of accidentally removing brain activity. Unlike ASR, which requires clean example data (user-provided or automatically determined), iCanClean does not depend on clean example data. iCanClean can theoretically be applied directly to contaminated data, thereby saving time (unnecessary data collection).

### 5.2. Main Findings

The empirical results of this phantom EEG study support our theoretical foundation for iCanClean. Unlike Adaptive Filtering, Auto-CCA, and ASR, our novel method provided consistently high Data Quality Scores, regardless of the type or number of artifacts present (i.e., for all conditions). Of particular note was the *Brain + All Artifacts* condition where iCanClean was the only algorithm to clean the data to satisfactory levels (i.e., similar score to data from the *Brain* condition). Thus, iCanClean shows promise as an all-in-one solution for removing a wide variety of artifacts.

### 5.3. Patterns Supporting the Theoretical Foundation of iCanClean

There are some patterns within the results worth noting that suggest ways to optimize artifact cleaning. First, there was a benefit to cleaning EEG data using mixtures of the EEG channels themselves. Whereas Adaptive Filtering cleans EEG data strictly using mixtures of reference noise signals, iCanClean can also use mixtures of the EEG channels, if desired. As anticipated, we saw improved performance when iCanClean was set to clean the corrupted EEG channels using mixtures of (raw) EEG channels themselves rather than using mixtures of reference (or pseudo-reference) noise channels. For example, iCanClean improved the *Brain + All Artifacts* condition to 55.9% using mixtures of EEG channels versus 39.0% using mixtures of noise channels. Second, there was a benefit to using CCA to identify common information between corrupt EEG data and reference noise signals prior to linear regression. Even when iCanClean was set to clean the corrupted EEG channels using only mixtures of noise channels (similar to Adaptive Filtering), iCanClean still achieved higher scores than Adaptive Filtering. For example, for the *Brain + All Artifacts* condition, iCanClean achieved a score of 39.0% using only mixtures of pseudo-reference noise channels, and Adaptive Filtering with the same pseudo-reference noise channels scored 32.9%. Third, there was a benefit to using reference noise signals to guide CCA toward finding noise components (i.e., the iCanClean approach) as opposed to applying CCA to a lagged copy of the EEG signals for blind separation (i.e., the Auto-CCA approach). Besides better Data Quality Scores in general, the advantage of iCanClean over Auto-CCA is readily evident in the number of components that were removed for the *Brain + Neck Muscle* condition. A total of four independent neck-muscle sources were sent into the phantom head. Therefore, in the ideal scenario, we expect a cleaning algorithm to identify exactly four distinct artifactual sources (or five with line noise). When 8-channel neck EMG data were available as the reference noise signals, iCanClean removed a total of 4 components and achieved a score of 56.4%. When no neck EMG signals were available, iCanClean removed 6 components and achieved a score of 58.1%. Meanwhile, Auto-CCA removed 43 components to achieve a score of 54.8%. This emphasizes that Auto-CCA may be more likely to remove brain activity, or to not remove artifacts at all, because Auto-CCA does not exploit reference noise signals like iCanClean does to direct CCA toward finding the underlying artifacts.

### 5.4. Other Patterns Worth Noting

Regarding using IMU signals to help remove EEG motion artifacts, as expected we saw improved performance when the base frequency of the filter bank matched the main frequency of the motion (0.95 Hz), but the effect was mild (37.8% at 0.95 Hz vs. 37.7 at 0.85 Hz; 4 harmonics and cascade filtering). We observed poor performance using an L_infinity_ solution (see [25]) as opposed to the traditional L_2_ solution for adaptive filtering (2.43% vs. 34.39%; 0.95 Hz, 1 harmonic). We also found that simultaneously fitting all reference IMU signals to the EEG channels with the traditional regression approach was better than the cascade approach proposed in [25] (41.4% vs. 37.8%; 0.95 Hz, 4 harmonics). Regarding ASR, we found its performance was improved with the use of an external calibration dataset as opposed to automatically determining clean subsections of the data to use for calibration (27.6% vs. 23.4 for *Brain + All Artifacts*).

### 5.5. Unexpected Results

There were also unexpected results. First, Auto-CCA did surprisingly well with eye artifact removal. Auto-CCA typically aims to remove low-correlation Auto-CCA components such as muscle artifacts. Here, for sake of completeness, we also tested removing high-correlation Auto-CCA components. Auto-CCA achieved good results removing high-correlation eye-blink artifacts at the optimal settings. Note, however, that Auto-CCA components containing brain activity will have a relatively high correlation, similar to eye artifacts. Users should be cautious when rejecting high-correlation components with the Auto-CCA approach. Second, we were surprised that the pseudo-reference version of iCanClean worked so well. We expected that dedicated noise sensors such as dual-layer EEG would better capture motion artifacts than pseudo-reference signals, but this may not always be the case. For the *Brain + Neck Muscles* condition, using neck EMG reference electrodes scored 56.4% while the pseudo-reference version of iCanClean scored 58.1%. For the *Brain + Walking Motion* condition, using dual-layer noise sensors scored 45.6% while the pseudo-reference version of iCanClean scored 54.3%.

### 5.6. Broader Implications of iCanClean

There are two main outcomes from this study. First, iCanClean increases the fidelity of EEG mobile brain-imaging studies done without accurate EEG noise sensors and suggests that some studies may not need noise sensors at all. Second, due to its minimal processing time, iCanClean would be particularly helpful for real-time brain–computer-interfaces in mobile applications (e.g., thought control of virtual reality, thought control of exoskeletons, and neurofeedback for physical rehabilitation).

On the first point, one downside to Adaptive Filtering is its reliance on accurate noise sensors. When saving cost is a major consideration, an inertial measurement unit (IMU) on the head is a popular choice of noise sensor [25,41,42,43,44]. IMU technology is cheap and readily available in a small form factor. Intuition suggests that signals from an IMU placed on the head should contain information about the motion artifacts of the electrodes placed on the head. However, in our experience, IMU signals are not strongly related to the motion artifacts that appear on EEG electrodes [6,45]. Note that motion artifacts are largely due to cable sway, which is not directly related to the movement of the head. Indeed, motion artifacts can be induced simply by moving the cables themselves, even if the head is stationary and the electrodes do not move relative to the scalp [14]. Considering the relatively poor quality of IMU signals as reference noise signals, additional modifications to Adaptive Filtering may need to be employed to remove motion artifacts. For example, in [25], the IMU-based algorithm employs narrow filter banks, a Volterra series expansion [46], an H-infinity update law [47], and cascade filtering [48]. We implemented an equivalent offline version of the IMU-based filtering algorithm in [25]. We found that the IMU-based algorithm did not provide suitable cleaning performance, despite the various modifications to Adaptive Filtering. Meanwhile, we showed that iCanClean can remove motion artifacts (and other artifacts in general), using only pseudo-reference signals (no dedicated noise sensors). Thus, iCanClean may advance mobile brain-imaging research by alleviating the need for accurate noise sensors and, perhaps, the need for noise sensors at all.

On the second point, iCanClean ran very quickly without extensive computational power. For the *Brain + All Artifacts* condition, iCanClean took 6 s to complete, whereas ASR, Auto-CCA, and Adaptive Filtering took 523, 6, and 87 s, respectively. Although we implemented the cleaning algorithms offline in the present manuscript, the iCanClean algorithm was computationally efficient and shows potential for real-time cleaning applications. iCanClean may prove valuable in providing an all-in-one clearing approach for mobile brain–computer interfaces by removing multiple types of artifacts, simultaneously, in a computationally efficient manner, without requiring dedicated hardware to record reference noise signals.

### 5.7. Study Limitations

There were some limitations to this study. First, it was not feasible to test every cleaning algorithm in the literature, so we limited our selection to ASR, Auto-CCA, and Adaptive Filtering. For example, we did not test the algorithm in [40], which used dual-layer EEG and a fast Fourier transform-based method to identify frequency bins for removal. However, we made it easy for others to validate our findings and test out their own algorithms by providing the data used in the experiment as well as the iCanClean algorithm as a downloadable plugin for EEGLAB. Included in the data are all the raw phantom EEG data and all MATLAB scripts used for processing and analysis. The second limitation to this study is we did not attempt to cancel eye blinks using electrooculogram (EOG) sensors on our phantom head apparatus. Our primary concern was to collect reference noise signals for the *Brain + Neck Muscle* condition. Future studies could incorporate EOG sensors on real humans using iCanClean to measure the efficacy of the approach, both at rest and for mobile tasks.

### 5.8. Recommendations for Practical Implementation of iCanClean

We have some recommendations for the practical implementation of iCanClean. First, when reference noise signals contain brain activity, average re-referencing to temporarily remove brain activity may generally be a good idea (see EEGLAB pop_reref function). For example, our neck muscle EMG recordings shared a reference electrode with the EEG system, so the raw EMG recordings naturally contain some brain activity. If CCA were used to search for latent relationships between the raw EMG and EEG recordings, it could remove brain activity (because brain activity would be in common to both the reference noise signals and the contaminated EEG). To average re-reference the EMG sensors to themselves, first calculate the average across all EMG channels (with their raw reference). Next, subtract this average value (time series) from each of the original EMG channels (forcing their new average to be 0). In this way, the average re-rereferencing can remove common activity across all EMG sensors, and one rank of data is deleted. Our results show this can help attenuate brain activity prior to CCA and improve performance (improved data quality scores) in this scenario. Second, to determine the best R^2^ threshold, we recommend new users view an optional plot with the iCanClean GUI that shows the R^2^ correlation as a function of the CCA component number (“Plot R^2^ correlation” checkbox in Appendix A). This should make it easier to more quickly determine the optimal range of R^2^ values for their particular setup (type of noise sensors, number of sensors, the task being studied, etc.). Third, users may want to vary the window size for their application. When using CCA to identify the noise components (File S1, Equation (1)), the user has a choice as to what sections of data (time windows) to include. The noise components can be calculated on the entire set of data or with a moving window. Assuming the data is stationary, longer windows should be better able to find and delete latent noise components by exploiting more total data. Meanwhile, shorter windows could better handle non-stationary data.

### 5.9. Recommendations for Future Research

Future research could focus on determining optimal parameter settings for iCanClean in various scenarios. In previous work, we showed that a moving-window version of iCanClean with 120 + 120 dual-layer EEG sensors can improve the ICA decomposition of motion-artifact data, for both table tennis [6] and uneven-terrain treadmill walking [49]. Based on [49], it appears that a shorter time window (2–4 s) is better for removing motion artifacts with dual-layer EEG in human data, as opposed to a longer window. We suspect this is due to nonstationarity in the data. Further research is needed to determine if performance can be improved by incorporating larger windows of data. If there proves to be a benefit to using more data, then it is possible we could implement CCA recursively in the future, given that recursive CCA algorithms exist in the literature [42,50,51,52]. This would reduce the computational cost of calculating CCA on a very large (and ever growing) window of data for real-time applications. The performance using a moving 2 s window is already relatively good [6,49]. This makes iCanClean easy to implement in real-time and minimizes computational cost. As a brief demonstration of iCanClean’s real-time cleaning capability, we applied iCanClean to remove artifacts from human data in pseudo-real time (see Appendix A). In future work, we would also like to attempt to remove other types of artifacts. For example, we believe there is significant potential for iCanClean to improve EEG studies that simultaneously record fMRI [53] or apply transcranial magnetic stimulation (TMS) [54], both of which induce large artifacts on the EEG recordings. For example, the work in [55,56,57,58] may be improved by applying iCanClean.

Finally, the last bit of future research we are excited to share is iCanClean’s potential to identify brain components from noisy EEG data, similar to ICA, but with less total data needed. Given the surprisingly good performance of the pseudo-reference version of iCanClean removing artifacts from channel-level data, we further explored the data, and we discovered that iCanClean was not only identifying noise sources for removal, but it was also identifying brain sources worth keeping. Additionally, iCanClean made the brain sources easy to identify based on their R^2^ sorting from canonical correlation analysis. The 10 ground-truth brain sources were exactly contained in the 10 highest numbered iCanClean components (the components that least resembled noise, i.e., those with the lowest R^2^ value). In Figure 11, we quantify iCanClean’s ability to identify brain components from the *Brain + All Artifacts* phantom dataset. Impressively, iCanClean recovered 6 (out of 10) brain sources with a score of at least 85%, and 9 (out of 10) brain sources were recovered scoring at least 75%. Based on these preliminary results, we decided to release an additional option to find candidate brain components with iCanClean as a plugin for EEGLAB (see Appendix A). We also briefly tested the concept on human data and saw promising results (finding candidate brain components using as little as 3 min of data with no preprocessing needed), but further systematic testing is needed.

## 6. Conclusions

iCanClean is a promising, all-in-one, real-time-capable cleaning algorithm which could facilitate the analysis of EEG data corrupted by a wide variety of artifacts. It is computationally efficient and can simultaneously remove multiple types of artifacts. We included relevant scripts and data associated with this work for download for validation by other research groups. We encourage others to test out the iCanClean algorithm on their data, and, similarly, we invite others to test out their own algorithms on the phantom EEG dataset with known ground-truth signals.

## 7. Patents

“Removing Latent Noise Components from Data Signals”, non-provisional patent application submitted to United States Patent and Trademark Office on 25 August 2021, Application No. PCT/US21/71283. Published under International Publication No. WO 2022/061322 on 24 March 2022. Claims priority to provisional patent application “A Novel Electroencephalography (EEG) Cleaning Algorithm that Uses Reference Noise Recordings and Canonical Correlation Analysis to Identify and Remove Artifacts”, previously submitted on 18 September 2020, Serial No. 63/080,475

“Using Pseudo Reference Noise Signals to Remove Latent Noise from Data Signals and Identify Data Sources”, provisional patent application submitted to United States Patent and Trademark Office on 5 September 2023, Serial No. 63/580,664.

## Figures and Tables

**Figure 1 sensors-23-08214-f001:**
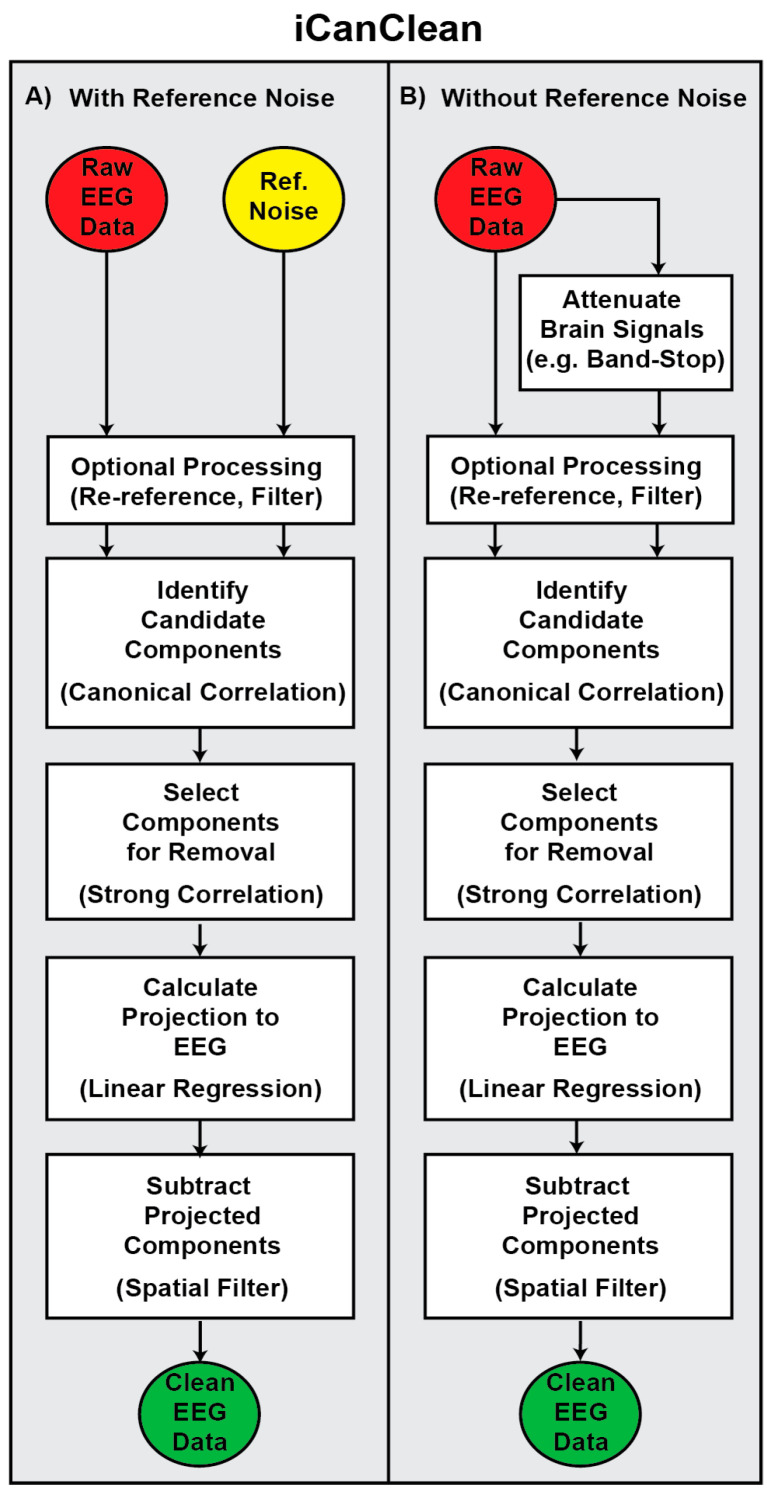
(**A**) The iCanClean algorithm uses canonical correlation analysis (CCA) to identify latent relationships between raw (noisy) EEG data and reference noise data (e.g., dual-layer noise sensor recordings). iCanClean uses CCA to identify subspaces (mixtures) of the EEG data that are correlated with subspaces of the noise data. iCanClean marks CCA components for removal based on which components have the strongest correlation. The user selects the correlation threshold and decides whether the noise components should be constructed from mixtures of the EEG and/or mixtures of the noise channels. iCanClean calculates the projection from the noise components onto the EEG data with linear regression. Finally, iCanClean directly subtracts the scaled noise components from the EEG channels. Because the noise components were calculated as a linear mixture of the EEG and/or noise channels and because the projection was calculated as a linear mixture of noise components, the iCanClean algorithm can be considered a spatial filter. (**B**) When reference noise signals cannot directly be recorded by dedicated hardware, iCanClean can optionally extract pseudo-reference signals from the raw EEG data by applying a temporal filter to attenuate brain activity. For example, a 5–45 Hz band-stop (notch) filter will significantly remove brain activity from the EEG data while mostly sparing the artifacts. After extracting pseudo-reference signals, the rest of the iCanClean algorithm is the same.

**Figure 2 sensors-23-08214-f002:**
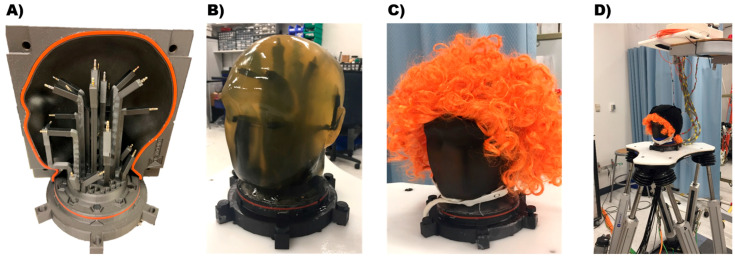
(**A**) We designed and 3D-printed an electrical phantom head mold for this study, based on the Open EEG Phantom project [34]. We placed 10 antennae inside the head to broadcast brain sources (ground-truth signals known). Additionally, 11 more antennae broadcast non-brain sources. Two were for eye blinks (electrically coupled to a single source); one was for eye saccades; four were for neck-muscle artifacts (bilateral trapezius and sternocleidomastoid); and four were for facial-muscle artifacts (bilateral masseter and temporalis). (**B**) We filled the mold with a mixture of ballistics gelatin, water, and salt to mimic the physical and electrical properties of the human head. (**C**) We placed conductive fabric (EeonTex LTT-PI-100, Marktek Inc., USA; not shown) and a wig over the phantom head to act as scalp and hair. (**D**) We secured the phantom head to a robotic platform (hexapod), which we used to induce walking motion artifacts. Dual-layer EEG electrodes were placed on the head so that reference motion artifact signals could be recorded alongside traditional EEG. After gelling the scalp-facing electrodes, the outward-facing (noise) electrodes were covered with conductive fabric (EeonTex LTT-PI-100) and kept electrically isolated from the scalp-facing electrodes. EEG amplifiers were secured above the phantom, with cables hanging loosely but bundled. Eight external sensors (not shown) were placed over the neck and connected to the EEG amplifier. These sensors recorded neck electromyography (EMG) which could be used as reference noise signals to remove neck-muscle artifacts. Similarly, the dual-layer EEG sensors recorded motion and line-noise artifacts.

**Figure 3 sensors-23-08214-f003:**
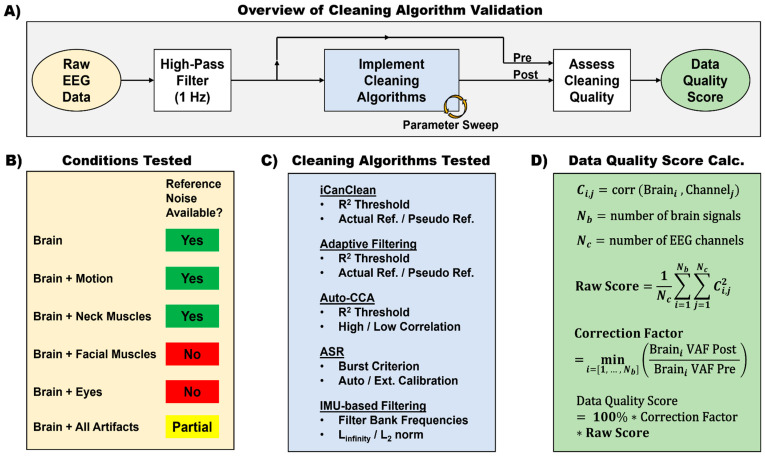
(**A**) Summary of the process used to validate iCanClean and compare it to other cleaning algorithms. Raw data were imported then high-pass filtered. We did not perform any channel rejection or re-reference the data. Minimally processed data were then sent through various cleaning algorithms, each with their own parameter sweep. After each preprocessing iteration, the cleaning effectiveness was quantified by a Data Quality Score, using ground-truth knowledge of the underlying brain sources. (**B**) Summary of the conditions that were tested and which conditions had reference noise recordings available that could be used to assist cleaning. (**C**) Summary of cleaning algorithms tested and the main parameters that were varied. (**D**) How the data quality score was calculated based on Pearson’s R^2^ correlation. The raw score was calculated by summing the R^2^ correlation across brain sources and taking the average correlation across EEG channels. A correction factor was then included to penalize any accidental deletion of brain activity. Since the ground-truth brain sources are known, we were able to calculate each brain source’s variance accounted for (VAF) via regression, i.e., the best that each brain source could possibly be reconstructed using a linear mixture of the EEG channels. By taking the minimum ratio of VAF Post to VAF Pre cleaning across all brain sources, we strongly penalized accidental deletion of any of the ground-truth brain sources.

**Figure 4 sensors-23-08214-f004:**
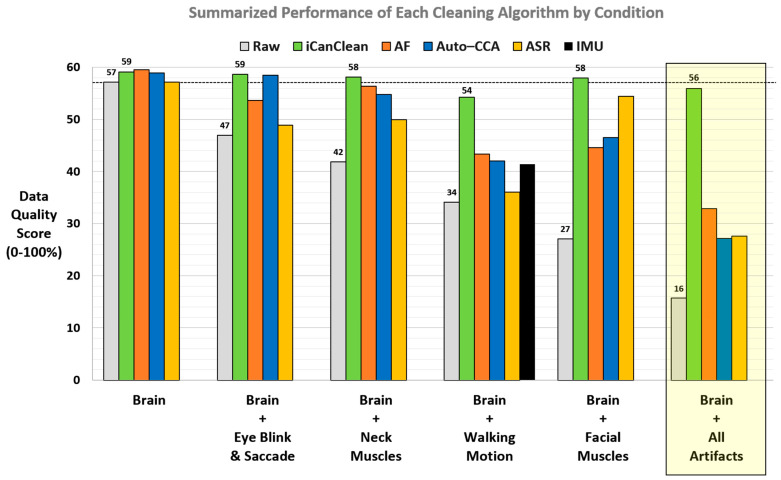
Results showing each cleaning method’s best Data Quality Score for each condition tested. Conditions are sorted from left to right in terms of their raw Data Quality Score (no cleaning, gray bar). The horizontal dashed line represents the Data Quality Score of the *Brain* condition prior to any cleaning (57%). This represents a reasonable target score when cleaning the *Brain + artifact* conditions (eye, neck, walking, facial, all). iCanClean (green bar) consistently outperformed the other cleaning methods and was the only method that could suitably clean the *Brain + All Artifacts* conditions.

**Figure 5 sensors-23-08214-f005:**
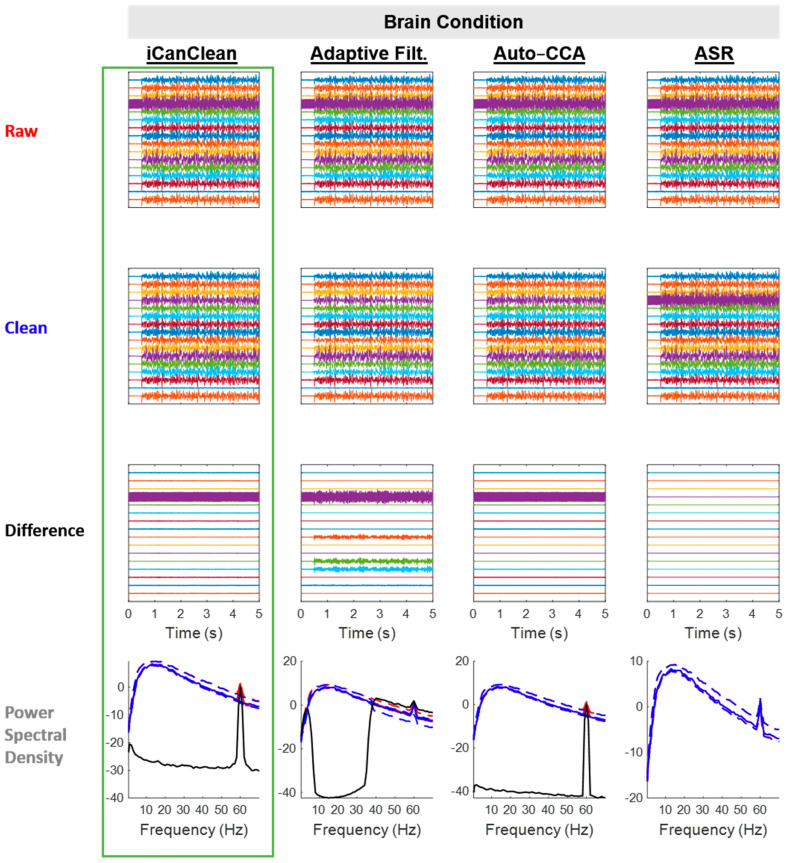
Qualitative (visual) assessment of the *Brain* condition before and after each processing method (iCanClean, Adaptive Filtering, Auto-CCA, ASR), using their respective ideal settings (maximized Data Quality Score after cleaning). Top row: time-series plot of raw EEG data for the *Brain* condition. Brain signals were sent to the phantom starting at time 0.5 s. Second row: ideally cleaned data. Third row: relative difference (what was removed during cleaning). Individual time-series plots in the Raw, Clean, and Difference plots are spaced 50 µV apart from each other, on center. Bottom row: power spectral density plots of the associated time-series data (Raw = red, Clean = blue, and Difference = black; solid lines = the median power across channels as a function of the frequency, dashed lines = the 25% and 75% percentiles).

**Figure 6 sensors-23-08214-f006:**
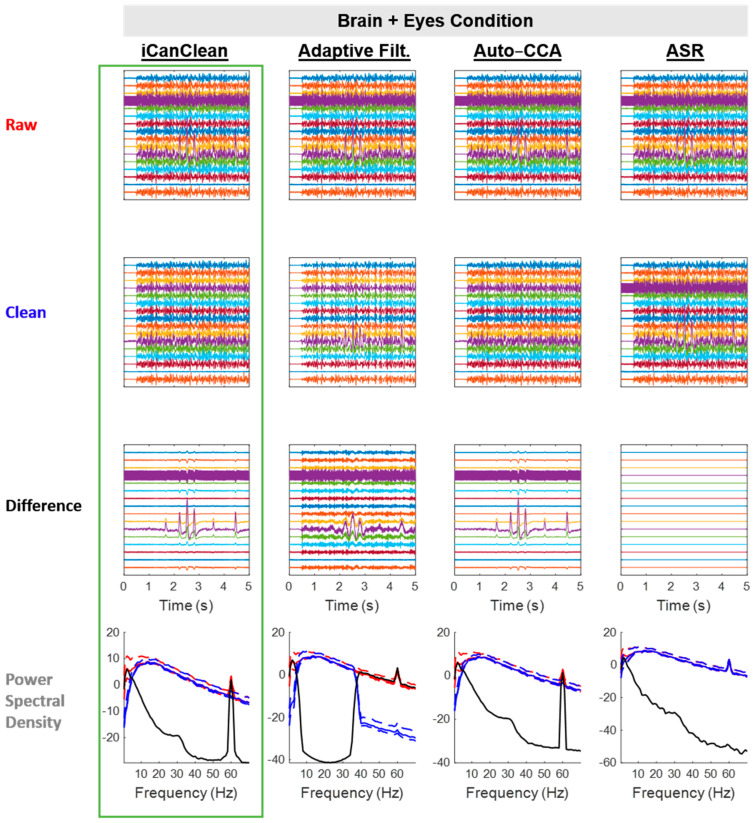
Qualitative (visual) assessment of the *Brain + Eyes* condition before and after each processing method (iCanClean, Adaptive Filtering, Auto-CCA, ASR), using their respective ideal settings (maximized Data Quality Score after cleaning). Top row: time-series plot of raw EEG data for the *Brain + Eyes* condition. Brain and eye signals were sent to the phantom starting at time 0.5 s. Second row: ideally cleaned data. Third row: relative difference (what was removed during cleaning). Individual time-series plots in the Raw, Clean, and Difference plots are spaced 50 µV apart from each other, on center. Bottom row: power spectral density plots of the associated time-series data (Raw = red, Clean = blue, and Difference = black; solid lines = the median power across channels as a function of the frequency, dashed lines = the 25% and 75% percentiles).

**Figure 7 sensors-23-08214-f007:**
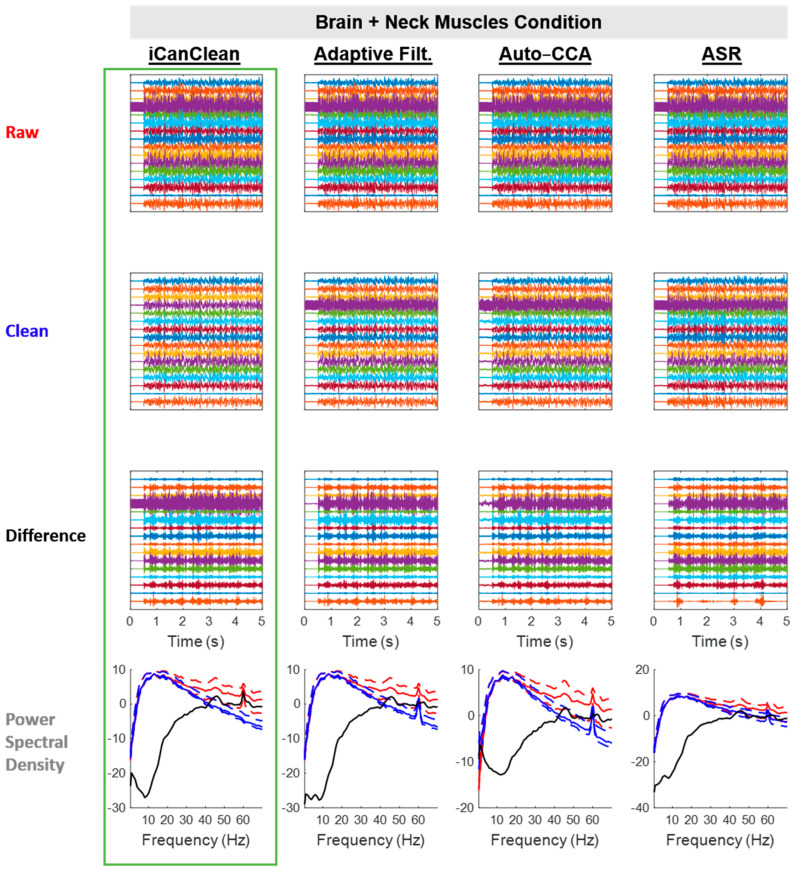
Qualitative (visual) assessment of the *Brain + Neck Muscles* condition before and after each processing method (iCanClean, Adaptive Filtering, Auto-CCA, ASR), using their respective ideal settings (maximized Data Quality Score after cleaning). Top row: time-series plot of raw EEG data for the *Brain + Neck Muscles* condition. Brain and neck-muscle signals were sent to the phantom starting at time 0.5 s. Second row: ideally cleaned data. Third row: relative difference (what was removed during cleaning). Individual time-series plots in the Raw, Clean, and Difference plots are spaced 50 µV apart from each other, on center. Bottom row: power spectral density plots of the associated time-series data (Raw = red, Clean = blue, and Difference = black; solid lines = the median power across channels as a function of the frequency, dashed lines = the 25% and 75% percentiles).

**Figure 8 sensors-23-08214-f008:**
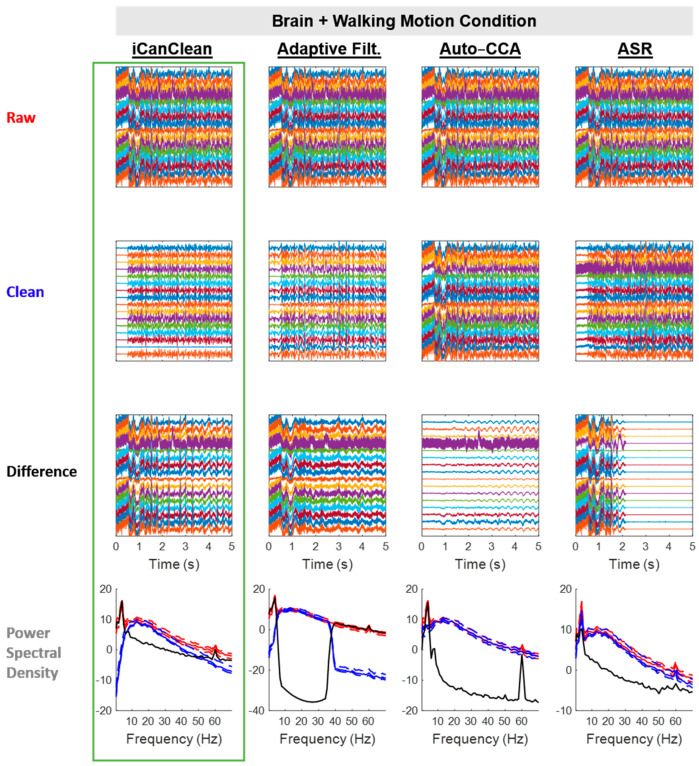
Qualitative (visual) assessment of the *Brain + Walking Motion* condition before and after each processing method (iCanClean, Adaptive Filtering, Auto-CCA, ASR), using their respective ideal settings (maximized Data Quality Score after cleaning). Top row: time-series plot of raw EEG data for the *Brain + Walking Motion* condition. Brain signals were sent to the phantom and the hexapod motion platform began to move at time 0.5 s. Second row: ideally cleaned data. Third row: relative difference (what was removed during cleaning). Individual time-series plots in the Raw, Clean, and Difference plots are spaced 50 µV apart from each other, on center. Bottom row: power spectral density plots of the associated time-series data (Raw = red, Clean = blue, and Difference = black; solid lines = the median power across channels as a function of the frequency, dashed lines = the 25% and 75% percentiles).

**Figure 9 sensors-23-08214-f009:**
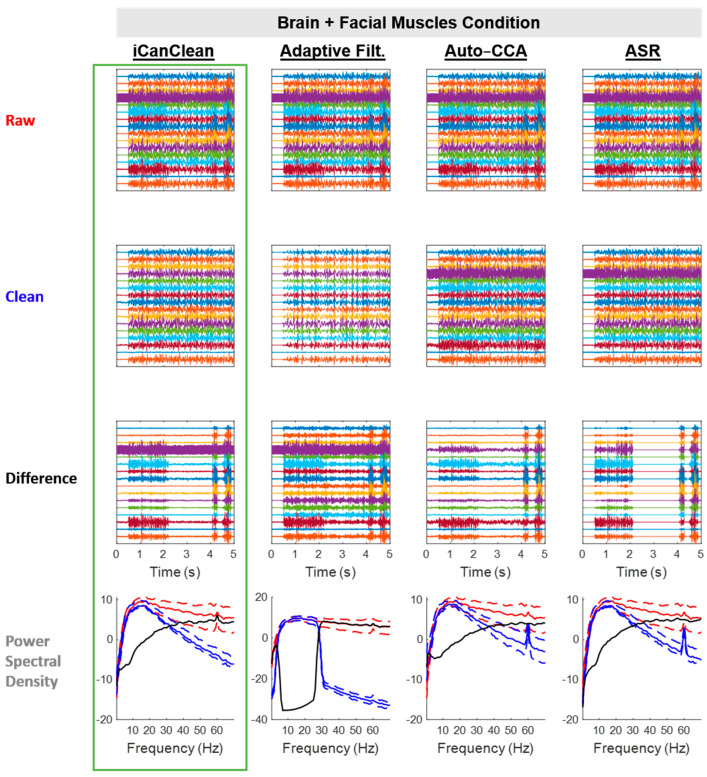
Qualitative (visual) assessment of the *Brain + Facial Muscles* condition before and after each processing method (iCanClean, Adaptive Filtering, Auto-CCA, ASR), using their respective ideal settings (maximized Data Quality Score after cleaning). Top row: time-series plot of raw EEG data for the *Brain + Facial Muscles* condition. Brain and facial-muscle signals were sent to the phantom starting at time 0.5 s. Second row: ideally cleaned data. Third row: relative difference (what was removed during cleaning). Individual time-series plots in the Raw, Clean, and Difference plots are spaced 50 µV apart from each other, on center. Bottom row: power spectral density plots of the associated time-series data (Raw = red, Clean = blue, and Difference = black; solid lines = the median power across channels as a function of the frequency, dashed lines = the 25% and 75% percentiles).

**Figure 10 sensors-23-08214-f010:**
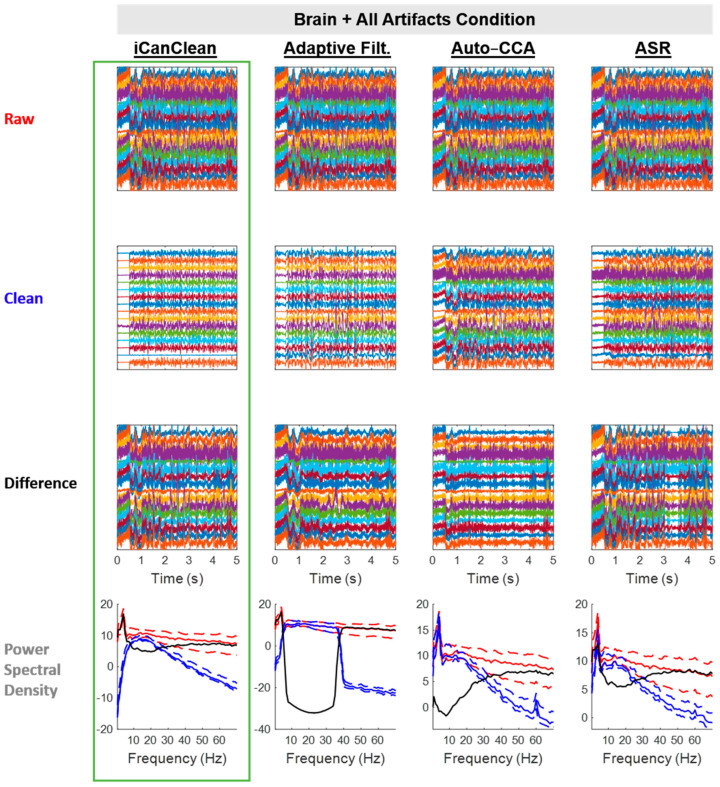
Qualitative (visual) assessment of the *Brain + All Artifacts* condition before and after each processing method (iCanClean, Adaptive Filtering, Auto-CCA, ASR), using their respective ideal settings (maximized Data Quality Score after cleaning). Top row: time-series plot of raw EEG data for the *Brain + All Artifacts* condition. Brain, eye, neck-muscle, and facial-muscle signals were sent to the phantom and the hexapod motion platform moved starting at time 0.5 s. Second row: ideally cleaned data. Third row: relative difference (what was removed during cleaning). Individual time-series plots in the Raw, Clean, and Difference plots are spaced 50 µV apart from each other, on center. Bottom row: power spectral density plots of the associated time-series data (Raw = red, Clean = blue, and Difference = black; solid lines = the median power across channels as a function of the frequency, dashed lines = the 25% and 75% percentiles).

**Figure 11 sensors-23-08214-f011:**
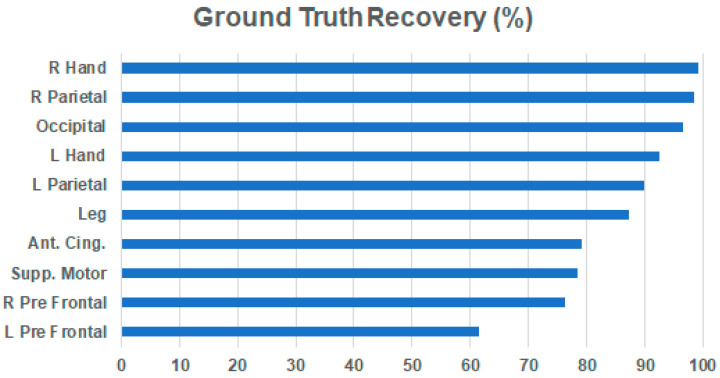
Ability of iCanClean to recover ground-truth brain sources from the *Brain + All Artifacts* condition. Here we used pseudo-reference signals in iCanClean (band stop 4–50 Hz), but instead of using iCanClean to clean the channels, we checked if any of the iCanClean components resembled the ground-truth brain components. Plotted is the R^2^ correlation between each ground-truth brain component (10 total, labeled by anatomical location in the phantom head) and the iCanClean component that best matches it. Note that these values have been normalized according to the best that any linear unmixing algorithm could do (e.g., ICA). For example, the best possible R^2^ correlation for the left hand is 63.1% (strictly limited to using linear mixtures of the EEG channels), and iCanClean found a candidate component with an R^2^ correlation of 58.3%. After normalizing, the ground-truth recovery value for this brain source is 92.4% (58.3/63.1 = 92.4%). Overall, iCanClean did surprisingly well as it reconstructed 6 out of 10 brain components with a score > 85% and 9 out of 10 > 75%. Identifying brain sources was not our original goal, but it helps explain why the pseudo-reference version performed so well. Note also that iCanClean auto sorted all the brain components to the end of the list. Out of 128 total components, the best matching components were #119-128 (i.e., the last 10).

## Data Availability

All relevant scripts and data are available for download on OpenNeuro (dataset number ds004784). https://openneuro.org/datasets/ds004784 (accessed on 29 August 2023).

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
