# Peer review of "iCanClean Removes Motion, Muscle, Eye, and Line-Noise Artifacts from Phantom EEG"

_sensors, 2023, doi:10.3390/s23198214_

Round 1

Reviewer 1 Report

The manuscript titled "A Novel Computational Approach (iCanClean) for Removing Artifacts from Scalp EEG Data Recorded in Mobile Conditions" presents a novel method, iCanClean, for removing various types of artifacts from scalp EEG data recorded in mobile conditions. The authors compare iCanClean with three other existing methods (Artifact Subspace Reconstruction, Auto-Canonical Correlation Analysis, and Adaptive Filtering) in the context of artifact removal from EEG data obtained from an electrical phantom head.

The study addresses a relevant and challenging problem in EEG research, as the presence of artifacts can significantly affect the quality and interpretability of EEG data, especially in mobile scenarios. The manuscript provides a clear and detailed description of the iCanClean algorithm and the experimental setup, including the construction of the electrical phantom head and the generation of ground-truth brain signals. The evaluation of iCanClean's performance under various artifact conditions is well-structured and informative.

However, there are several aspects that need to be addressed to improve the manuscript:

1.     Clarity of Abstract: The abstract could be more concise and structured. It should briefly mention the key findings and conclusions of the study. The current abstract is quite detailed and may overwhelm readers with technical information.

2.     Introduction: The introduction provides a good background on EEG and the challenges posed by artifacts. However, it could be made more concise. The information about the advantages of EEG in mobile scenarios seems somewhat extensive for an introduction.

3.     The section titled "The iCanClean Algorithm" is informative, but it is quite long and may benefit from some simplification for readability.

4.     The description of the construction of the electrical phantom head and the generation of ground-truth brain signals could be summarized more concisely. The current level of detail may be more appropriate for a methods section in a full research paper.

5.      How does the choice of reference noise signals (e.g., EEG sensors, neck muscle electromyography, or pseudo-reference signals) impact the efficacy of the iCanClean algorithm in removing various types of artifacts in EEG data, and what are the underlying mathematical principles that govern this impact?

6.     The presentation of results is thorough and detailed, which is commendable. However, the manuscript could benefit from a more concise presentation of results, focusing on the key findings and comparisons.

7.     The discussion section could be more focused on the implications of the results and the significance of iCanClean in the context of EEG artifact removal. It currently contains a lot of technical details that might be better placed in the methods section or supplementary materials.

8.     Data Quality Scores: The study mentions that iCanClean provided consistently high Data Quality Scores compared to other algorithms. Can you provide more details on how these scores are calculated? What specific metrics or criteria were used to assess data quality?

9.     Parameter Sensitivity: It's mentioned that the study didn't test all possible combinations of parameters for each cleaning algorithm. Could you elaborate on which parameters were explored for iCanClean and how their settings were determined? Were there any sensitivity analyses conducted to assess the robustness of the method to parameter choices?

10.  Real-time Application: The study suggests that iCanClean could be valuable for real-time brain-computer interfaces. Can you clarify how the method's computational efficiency was evaluated? Were there any latency measurements conducted for real-time applications?

11.  Artifact Types: The study discusses the removal of various artifact types, such as motion artifacts and eye blink artifacts. Can you provide insights into how iCanClean performs when multiple artifact types are present simultaneously in EEG data? Were there any scenarios where it struggled to remove specific types of artifacts effectively?

12.  Re-referencing: The study mentions average re-referencing to temporarily remove brain activity from reference noise signals. Could you explain in more detail how this re-referencing was implemented and its impact on the cleaning performance?

13.  Recursive CCA: The study suggests the possibility of implementing recursive CCA in the future to reduce computational costs. Can you provide more information about how this would work and the potential advantages it might offer for real-time applications?

14.  Identification of Brain Components: The study mentions that iCanClean can identify brain components from noisy EEG data. Could you explain how this identification process works and its potential applications in EEG research? Were there any limitations or challenges encountered when identifying brain components?

15.  Comparative Data: It would be helpful to have a summary or table that compares the performance of iCanClean against other popular cleaning algorithms across different conditions and types of artifacts. This could provide a clearer overview of iCanClean's strengths and weaknesses.

16.  Future Research: Could you elaborate on the specific future research directions or experiments that you recommend based on the findings of this study? How might the iCanClean method be further improved or extended for different EEG applications?

17.  Figures and Tables: The manuscript should include figures and tables that visually summarize the key results and comparisons. Visual aids can help readers understand the findings more easily.

It's satisfactory but minor revision is needed.

Author Response

Please see attached Word file with our response to Reviewer 1.

Reviewer 2 Report

The authors proposed a novel algorithm that can effectively and efficiently remove a range of artifacts present in EEG data, but there are a few concerns that I hope the authors could address.

- Line 38, "EEG also has the ability to localize brain sources with reasonable spatial resolution." It would be beneficial if the authors could use some citation here and be more specific about "reasonable resolution", e.g. xx centimeters

- Line 68. The ICA computation time varies greatly depending on the length of the recording, the number of channels and the algorithm used. The authors should be specific about under which circumstances does ICA require multiple hours, and would be best if substantiated with citations. 

- Line 70-72, "Our general recommendation is to record at least 30 minutes of high-density EEG (100+ channels) at a sampling frequency of at least 500 Hz when attempting to separate sources with ICA." The authors should more rigorously justify the recommendation by commenting on how these numbers are derived or using a citation

- Line 203, the authors stated that preprocessing of data is optional. Could the authors briefly comment on how preprocessing may affect the performance of the algorithm, and if preprocessing does affect the final results, what would be the recommended preprocessing procedures?

- Line 383-385, " ...(0-100%, higher is better). Note that a score of 100% is achieved when the time series of the EEG channels are strictly mixtures of the 10 brain sources and nothing else (no other extraneous sources to introduce noise)." I don't think this is true. One can disprove this using a simple simulation in Matlab: suppose there are 2 brain sources each containing 100 data points, x1=randn(100,1) and x2=randn(100,2), and only one channel which happens to be a 50-50 mixture of the two sources, i.e. y=0.5*x1+0.5*x2. According to the definition in the paper, the score would be score=corr(x1,y)^2+corr(x2,y)^2. This will NOT be 1 as the authors claim in this sentence. This should be revised, unless a rigorous mathematical proof can be provided.

- In all the power spectral density plots, what do the dashed lines stand for?

- Line 546, "...which should theoretically lead to better cleaning performance". It is not intuitive enough why this should be the case. The authors should either give a proof or tone it down

- Line 661, "For example, the IMU-based algorithm took 494 hours to clean 5 minutes of Brain + Walking Motion data, and it worsened the Data Quality Score..." It sounds like the parameters used in the algorithm may not be optimal. If so, I think it would be better comparison if the authors can compare the computational complexity when all the algorithms are using the optimal parameters. In the subsequent sentence where the authors detailed the chosen parameters, it is unclear what they mean to a reader who is not familiar with the algorithm. The authors should consider making the information more digestible.

- Line 861, "The next noise component pair (U2 , V2) has the second largest R2 correlation and is independent of the first component pair" It is unclear what the authors are referring to by stating the component pairs are independent. This should be better clarified, as U1 and U2 (or V1 and V2) are clearly not independent. Suppose X=[x1, x2] where x1 and x2 are column vectors, U1=a*x1+b*x2, and U2=c*x1+d*x2. cov(U1,U2)=ac*var(x1)+bd*var(x2)+(ad+bc)*cov(x1,x2) and this is in general non-zero, indicating dependency.

Author Response

Please see attached Word document for our response to Reviewer 2.

Round 2

Reviewer 1 Report

I appreciate authors for revision and all concerns were addressed!